# Faster Boosting with Smaller Memory

**Julaiti Alafate**
Department of Computer Science and Engineering
University of California, San Diego
La Jolla, CA 92093
jalafate@eng.ucsd.edu

**Yoav Freund**
Department of Computer Science and Engineering
University of California, San Diego
La Jolla, CA 92093
yfreund@ucsd.edu

## Abstract

State-of-the-art implementations of boosting, such as XGBoost and LightGBM, can process large training sets extremely fast. However, this performance requires that the memory size is sufficient to hold a 2-3 multiple of the training set size. This paper presents an alternative approach to implementing the boosted trees, which achieves a significant speedup over XGBoost and LightGBM, especially when the memory size is small. This is achieved using a combination of three techniques: early stopping, effective sample size, and stratified sampling. Our experiments demonstrate a 10-100 speedup over XGBoost when the training data is too large to fit in memory.

## 1 Introduction

Boosting [7, 16], and in particular gradient boosted trees [9], are some of the most popular learning algorithms used in practice. There are several highly optimized implementations of boosting, among which **XGBoost** [5] and **LightGBM** [12] are two of the most popular ones. These implementations can train models with hundreds of trees using millions of training examples in a matter of minutes.

However, a significant limitation of these methods is that all of the training examples are required to be stored in the main memory. For LightGBM this requirement is strict. XGBoost can operate in the disk-mode, which makes it possible to use machines with smaller memory than the training set size. However, it comes with a penalty in much longer training time.

In this paper, we present a new implementation of boosted trees[1]. This implementation can run efficiently on machines whose memory sizes are much smaller than the training set. It is achieved with no penalty in accuracy, and with a speedup of 10-100 over XGBoost in disk mode.

Our method is based on the observation that each boosting step corresponds to an estimation of the gradients along the axis defined by the weak rules. The common approach to performing this estimation is to scan *all* of the training examples so as to minimize the estimation error. This operation is very expensive especially when the training set does not fit in memory.

We reduce the number of examples scanned in each boosting iteration by combining two ideas. First, we use *early stopping* [19] to minimize the number of examples scanned at each boosting iteration.

Second, we keep in memory only a sample of the training set, and we replace the sample when the sample in memory is a poor representation of the complete training set. We exploit the fact that boosting tends to place large weights on a small subset of the training set, thereby reducing the effectiveness of the memory-resident training set. We propose a measure for quantifying the variation in weights called the *effective number of examples*. We also describe an efficient algorithm, *stratified weighted sampling*.

Early stopping for Boosting was studied in previous work [6, 4]. The other two are, to the best of our knowledge, novel. In the following paragraphs, we give a high-level description of these three ideas, which will be elaborated on in the rest of the paper.

**Early Stopping**    We use early stopping to reduce the number of examples that the boosting algorithm reads from the memory to the CPU. A boosting algorithm adds a weak rule to the combined strong rule iteratively. In most implementations, the algorithm searches for the *best* weak rule, which requires scanning *all* of the training examples. However, the theory of boosting requires the added weak rule to be just *significantly better than random guessing*, which does not require scanning *all* of the training examples. Instead, our approach is to read just as many examples as needed to identify a weak rule that is significantly better than random guessing.

Our approach is based on *sequential analysis* and *early stopping* [19]. Using sequential analysis methods, we designed a stopping rule to decide when to stop reading more examples without increasing the chance of over-fitting.

**Effective Number of Examples**    Boosting assigns different weights to different examples. The weight of an example represents the magnitude of its "influence" on the estimate of the gradient. However, when the weight distribution of a training set is dominated by a small number of "heavy" examples, the variance of the gradient estimates is high. It leads to over-fitting, and effectively reduces the size of the training set. We quantify this reduction using the *effective number of examples*, $n_{\text{eff}}$. To get reliable estimates, $n_{\text{eff}}$ should be close to the size of the current training set in memory, $n$. When $\frac{n_{\text{eff}}}{n}$ is small, we flush the current training set, and get a new sample using *weighted sampling*.

**Stratified Weighted Sampling**    While there are well-known methods for weighted sampling, all of the existing methods (that we know of) are inefficient when the weights are highly skewed. In such cases, most of the scanned examples are rejected, which leads to very slow sampling. To increase the sampling efficiency, we introduce a technique we call *stratified weighted sampling*. It generates the same sampled distribution while guaranteeing that the fraction of rejected examples is no large than $\frac{1}{2}$.

We implemented a new boosted tree algorithm with these three techniques, called **Sparrow**. We compared its performance to the performance of **XGBoost** and **LightGBM** on two large datasets: one with 50 million examples (the human acceptor splice site dataset [18, 1]), the other with over 600 million examples (the bathymetry dataset [11]). We show that **Sparrow** can achieve 10-20x speed-up over **LightGBM** and **XGBoost** especially in the limited memory settings.

The rest of the paper is organized as follows. In Section 2 we discuss the related work. In Section 3 we review the confidence-based boosting algorithm. In Section 4 we describe the statistical theory behind the design of Sparrow. In Section 5 we describe the design of our implementation. In Section 6 we describe our experiments. We conclude with the future work direction in Section 7.

## 2   Related Work

There are several methods that use sampling to reduce the training time of boosting. Both Friedman *et al.* [8] and LightGBM [12] use a fixed threshold to filter out the light-weight examples: the former discards the examples whose weights are smaller than the threshold; the later accepts all examples if their gradients exceed the threshold, otherwise accepts them with a fixed probability. Their major difference with Sparrow is that their sampling methods are biased, while Sparrow does not change the original data distribution. Appel *et al.* [2] uses small samples to prune weak rules associated with unpromising features, and only scans all samples for evaluating the remaining ones. Their major difference with Sparrow is that they focus on finding the "best" weak rule, while Sparrow tries to find

a "statistically significant" one. Scanning over all example is required for the former, while using a stopping rule the algorithm often stops after reading a small fraction of examples.

The idea of accelerating boosting with stopping rules is also studied by Domingo and Watanabe [6] and Bradley and Schapire [4]. Our contribution is in using a tighter stopping rule. Our stopping rule is tighter because it takes into account the dependence on the variance of the sample weights.

There are several techniques that speed up boosting by taking advantage of the sparsity of the dataset [5, 12]. We will consider those techniques in future work.

## 3 Confidence-Rated Boosting

We start with a brief description of the confidence-rated boosting algorithm under the AdaBoost framework (Algorithm 9.1 on the page 274 of [16]).

Let $\vec{x} \in X$ be the feature vectors and let the output be $y \in Y = \{-1, +1\}$. For a joint distribution $\mathcal{D}$ over $X \times Y$, our goal is to find a classifier $c : X \to Y$ with small error:

$$\text{err}_{\mathcal{D}}(c) \doteq P_{(\vec{x},y)\sim\mathcal{D}}\left[c(\vec{x}) \neq y\right].$$

We are given a set $\mathcal{H}$ of base classifiers (weak rules) $h : X \to [-1, +1]$. We want to generate a *score function*, which is a *weighted* sum of $T$ rules from $\mathcal{H}$:

$$S_T(\vec{x}) = \left(\sum_{t=1}^{T} \alpha_t h_t(\vec{x})\right).$$

The term $\alpha_t$ is the weights by which each base classifiers contribute to the final prediction, and is decided by the specific boosting paradigm.

Finally, we have the strong classifier as the sign of the score function: $H_T = \text{sign}(S_T)$.

AdaBoost can be viewed as a coordinate-wise gradient descent algorithm [15]. The algorithm iteratively finds the direction (weak rule) which maximizes the decrease of the average potential function, and then adds this weak rule to $S_T$ with a weight that is proportional to the magnitude of the gradient. The potential function used in AdaBoost is $\Phi(\vec{x}, y) = e^{-S_T(\vec{x})y}$. Other potential functions have been studied (e.g. [9]). In this work we focus on the potential function used in AdaBoost.

We distinguish between two types of average potentials: the expected potential or true potential:

$$\Phi(S_T) = E_{(\vec{x},y)\sim\mathcal{D}}\left[e^{-S_T(\vec{x})y}\right],$$

and the average potential or empirical potential:

$$\widehat{\Phi}(S_T) = \frac{1}{n}\sum_{i=1}^{n} e^{-S_T(\vec{x}_i)y_i}.$$

The ultimate goal of the boosting algorithm is to minimize the expected potential, which determines the true error rate. However, most boosting algorithms, including **XGBoost** and **LightGBM**, focus on minimizing the empirical potential $\widehat{\Phi}(S_T)$, and rely on the limited capacity of the weak rules to guarantee that the true potential is also small. **Sparrow** takes a different approach. It uses an estimator of the true edge (explained below) to identify the weak rules that reduce the *true* potential with high probability.

Consider adding a weak rule $h_t$ to the score function $S_{t-1}$, we get $S_t = S_{t-1} + \alpha_t h_t$. Taking the partial derivative of the average potential $\Phi$ with respect to $\alpha_t$ we get

$$\left.\frac{\partial}{\partial\alpha_t}\right|_{\alpha_t=0} \Phi(S_{t-1} + \alpha_t h) = E_{(\vec{x},y)\sim\mathcal{D}_{t-1}}\left[h(\vec{x})y\right] \tag{1}$$

where

$$\mathcal{D}_{t-1} = \frac{\mathcal{D}}{Z_{t-1}}\exp\left(-S_{t-1}(\vec{x})y\right), \tag{2}$$

and $Z_{t-1}$ is a normalization factor that makes $\mathcal{D}_{t-1}$ a distribution.

Boosting algorithms performs coordinate-wise gradient descent on the average potential where each coordinate corresponds to one weak rule. Using equation (1), we can express the gradient with respect to the weak rule $h$ as a correlation, which we call the *true edge*:

$$\gamma_t(h) \doteq \mathrm{corr}_{\mathcal{D}_{t-1}}(h) \doteq E_{(\vec{x},y)\sim\mathcal{D}_{t-1}}\left[h(\vec{x})y\right], \tag{3}$$

which is not directly measurable. Given $n$ i.i.d. samples, an unbiased estimate for the true edge is the *empirical edge*:

$$\hat{\gamma}_t(h) \doteq \widehat{\mathrm{corr}}_{\mathcal{D}_{t-1}}(h) \doteq \sum_{i=1}^{n} \frac{w_i}{Z_{t-1}} h(\vec{x}_i)y_i, \tag{4}$$

where $w_i = e^{-S_{t-1}(\vec{x}_i)}$ and $Z_{t-1} = \sum_{i=1}^{n} w_i$.

# 4 Theory

To decrease the expected potential, we want to find a weak rule with a large edge (and add it to the score function). XGBoost and LightGBM do this by searching for the weak rule with the largest *empirical edge*. Sparrow finds a weak rule which, with high probability, has a significantly large *true edge*. Next, we explain the statistical techniques for identifying such weak rules while minimizing the number of examples needed to compute the estimates.

## 4.1 Effective Number of Examples

Equation 4 defines $\hat{\gamma}(h)$, which is an unbiased estimate of $\gamma(h)$. How accurate is this estimate?

A standard quantifier is the variance of the estimator. Suppose the true edge of a weak rule $h$ is $\gamma$. Then the expected (normalized) correlation between the predictions of $h$ and the true labels, $\frac{w}{Z}yh(x)$, is $2\gamma$. The variance of this correlation can be written as $\frac{1}{n^2}\frac{E(w^2)}{E^2(w)} - 4\gamma^2$. Ignoring the second term (because $\gamma$ is usually close to zero) and the variance in $E(w)$, we approximate the variance in the edge to be

$$\mathrm{Var}(\hat{\gamma}) \approx \frac{\sum_{i=1}^{n} w_i^2}{\left(\sum_{i=1}^{n} w_i\right)^2}. \tag{5}$$

If all of the weights are equal then $\mathrm{Var}(\hat{\gamma}) = 1/n$. It corresponds to a standard deviation of $1/\sqrt{n}$ which is the expected relation between the sample size and the error.

If the weights are not equal then the variance is larger and thus the estimate is less accurate. We define the *effective number of examples* $n_{\mathrm{eff}}$ to be $1/\mathrm{Var}(\hat{\gamma})$, specifically,

$$n_{\mathrm{eff}} \doteq \frac{\left(\sum_{i=1}^{n} w_i\right)^2}{\sum_{i=1}^{n} w_i^2}. \tag{6}$$

To see that the name "effective number of examples" makes sense, consider $n$ weights where $w_1 = \cdots = w_k = 1/k$ and $w_{k+1} = \cdots = w_n = 0$. It is easy to verify that in this case $n_{\mathrm{eff}} = k$ which agrees with our intuition, namely the examples with zero weights do not affect the estimate.

Suppose the memory is only large enough to store $n$ examples. If $n_{\mathrm{eff}} \ll n$ then we are wasting valuable memory space on examples with small weights, which can significantly increase the chance of over-fitting. We can fix this problem by using weighted sampling. In this way we repopulate memory with $n$ equally weighted examples, and make it possible to learn without over-fitting.

## 4.2 Weighted Sampling

When **Sparrow** detects that $n_{\mathrm{eff}}$ is much smaller than the memory size $n$, it clears the memory and collects a new sample from disk using weighted sampling.

The specific sampling algorithm that **Sparrow** uses is the *minimal variance weighted sampling* [13]. This method reads from disk one example $(\vec{x}, y)$ at a time, calculates the weight for that example

$w_i$, and accepts the example with the probability proportional to its weight. Accepted examples are stored in memory with the initial weights of 1. This increases the effective sample size from $n_{\text{eff}}$ back to $n$, thereby reduces the chance of over-fitting.

To gain some intuition regarding this effect, consider the following setup of an imbalanced classification problem. Suppose that the training set size is $N = 100,000$, of which $0.01N$ are positive and $0.99N$ are negative. Suppose we can store $n = 2,000$ examples in memory. The number of the memory-resident examples is $0.01n = 20$. Clearly, with such a small number of positive examples, there is a danger of over-fitting. However, an (almost) all negative rule is 99% correct. If we then reweigh the examples using the AdaBoost rule, we will give half of the total weight to the positives and the other half to the negatives. The value of $n_{\text{eff}}$ will drop to about 80. This would trigger a resampling step, which generates a training set with 1000 positives and 1000 negatives. It allows us to find additional weak rules with little danger of over-fitting.

This process continues as long as **Sparrow** is making progress and the weights are becoming increasingly skewed. When the skew is large, $n_{\text{eff}}$ is small and **Sparrow** resamples a new sample with uniform weights.

Sparrow uses weighted sampling to achieve high disk-to-memory efficiency. In addition, Sparrow also achieves high memory-to-CPU efficiency by reading from memory the minimal number of examples necessary to establish that a particular weak rule has a significant edge. This is done using *sequential analysis* and *early stopping*.

## 4.3 Sequential Analysis

Sequential analysis was introduced by Wald in the 1940s [19]. Suppose we want to estimate the expected loss of a model. In the standard large deviation analysis, we assume that the loss is bounded in some range, say $[-M, +M]$, and that the size of the training set is $n$. This implies that the standard deviation of the training loss is at most $M/\sqrt{n}$. To make this standard deviation smaller than some $\epsilon > 0$, we need that $n > (M/\epsilon)^2$. While this analysis is optimal in the worst case, it can be improved if we have additional information about the standard deviation. We can glean such information from the observed losses by using the following sequential analysis method.

Instead of choosing $n$ ahead of time, the algorithm computes the loss one example at a time. It uses a *stopping rule* to decide whether, conditioned on the sequence of losses seen so far, the difference between the average loss and the true loss is smaller than $\epsilon$ with large probability. The result is that when the standard deviation is significantly smaller than $M/\sqrt{n}$, the number of examples needed in the estimate is much smaller than $(M/\epsilon)^2$.

We use a stopping rule based on Theorem 1 in Appendix B, which depends on both the mean and the variance of the weighted correlation [3]. Fixing the current strong rule $H$ (i.e. the score function), we define a (unnormalized) weight for each example, denoted as $w(\vec{x}, y) = e^{-H(x)y}$. Consider a particular candidate weak rule $h$ and a sequence of labeled examples $\{(\vec{x}_1, y_1), (\vec{x}_2, y_2), \ldots\}$. For some $\gamma > 0$, we define two cumulative quantities (after seeing $n$ examples from the sequence):

$$M_t \doteq \sum_{i=1}^{n} w(\vec{x}_i, y_i)(h_t(\vec{x}_i)y_i - \gamma), \text{ and } V_t \doteq \sum_{i=1}^{n} w(\vec{x}_i, y_i)^2. \tag{7}$$

$M_t$ is an estimate of the difference between the true correlation of $h$ and $\gamma$. $V_t$ quantifies the variance of this estimate.

The goal of the stopping rule is to identify a weak rule $h$ whose true edge is larger than $\gamma$. It is defined to be $t > t_0$ and

$$M_t > C\sqrt{V_t(\log\log \frac{V_t}{M_t} + B)}, \tag{8}$$

where $t_0, C$, and $B$ are parameters. If both conditions of the stopping rule are true, we claim that the true edge of $h$ is larger than $\gamma$ with high probability. The proof of this test can be found in [3].

Note that our stopping rule is correlated with the cumulative variance $V_t$, which is basically the same as $1/n_{\text{eff}}$. If $n_{\text{eff}}$ is large, say $n_{\text{eff}} = n$ when a new sample is placed in memory, the stopping rule stops quickly. On the other hand, when the weights diverge, $n_{\text{eff}}$ becomes smaller than $n$, and the stopping rule requires proportionally more examples before stopping.

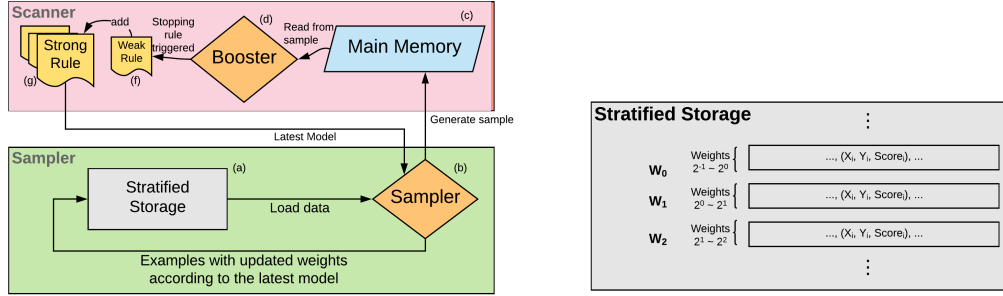

Figure 1: The **Sparrow** system architecture. Left: The workflow of the Scanner and the Sampler. Right: Partitioning of the examples stored in disk according to their weights.

The relationship between martingales, sequential analysis, and stopping rules has been studied in previous work [19]. Briefly, when the edge of a rule is smaller than $\gamma$, then the sequence is a supermartingale. If it is larger than $\gamma$, then it is a submartingale. The only assumption is that the examples are sampled i.i.d.. Theorem 1 in Appendix B guarantees two things about the stopping rule defined in Equation 8: (1) if the true edge is smaller than $\gamma$, the stopping rule will never fire (with high probability); (2) if the stopping rule fires, the true edge of the rule $h$ is larger than $\gamma$.

## 5 System Design and Algorithms

In this section we describe **Sparrow**. As Sparrow consists of a number of concurrent threads and many queues, we chose to implement it using the **Rust** programming language for the benefits of its memory-safety and thread-safety guarantees [14].

We use a bold letter in parenthesis to refer the corresponding component in the workflow diagram in Figure 1. We also provide the pseudo-code in the Appendix C.

The main procedure of **Sparrow** generates a sequence of weak rules $h_1, \ldots, h_k$ and combines them into a strong rule $H_k$. It calls two subroutines that execute in parallel: a **Scanner** and a **Sampler**.

**Scanner** The task of a scanner (the upper part of the workflow diagram in Figure 1) is to read training examples sequentially and stop when it has identified one of the rules to be a *good* rule.

At any point of time, the scanner maintains the current strong rule $H_t$, a set of candidate weak rules $\mathcal{W}$, and a target edge $\gamma_{t+1}$. For example, when training boosted decision trees, the scanner maintains the current strong rule $H_t$ which consists of a set of decision trees, a set of candidate weak rules $\mathcal{W}$ which is the set of candidate splits on all features, and $\gamma_{t+1} \in (0.0, 0.5)$.

Inside the scanner, a booster **(d)** scans the training examples stored in main memory **(c)** sequentially, one at a time. It computes the weight of the read examples using $H_t$ and then updates a running estimate of the edge of each weak rule $h \in \mathcal{W}$ accordingly. Periodically, it feeds these running estimates into the stopping rule, and stop the scanning when the stopping rule fires.

The stopping rule is designed such that if it fires for $h_t$, then the true edge of a particular weak rule $\gamma(h_{t+1})$ is, with high probability, larger than the set threshold $\gamma_{t+1}$. The booster then adds the identified weak rule $h_{t+1}$ **(f)** to the current strong rule $H_t$ to create a new strong rule $H_{t+1}$ **(g)**. The booster decides the weight of the weak rule $h_{t+1}$ in $H_{t+1}$ based on $\gamma_{t+1}$ (lower bound of its accuracy). It could underestimate the weight. However, if the underestimate is large, the weak rule $h_{t+1}$ is likely to be "re-discovered" later which will effectively increase its weight.

Lastly, the scanner falls into the *Failed* state if after exhausting all examples in the current sample set, no weak rule with an advantage larger than the target threshold $\gamma_{t+1}$ is detected. When it happens, the scanner shrinks the value of $\gamma_{t+1}$ and restart scanning. More precisely, it keeps track of the empirical edges $\hat{\gamma}(h)$ of all weak rules $h$. When the failure state happens, it resets the threshold $\gamma_{t+1}$ to just below the value of the current maximum empirical edge of all weak rules.

To illustrate the relationship between the target threshold and the empirical edge of the detected weak rule, we compare their values in Figure 2. The empirical edge $\hat{\gamma}(h_{t+1})$ of the detected weak

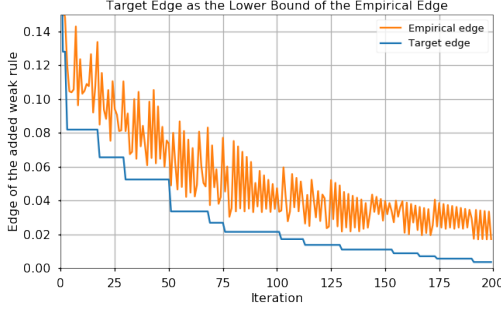

Figure 2: The empirical edge and the corresponding target edge $\gamma$ of the weak rules being added to the ensemble. Sparrow adds new weak rules with a weight calculated using the value of $\gamma$ at the time of their detection, and shrinks $\gamma$ when it cannot detect a rule with an edge over $\gamma$.

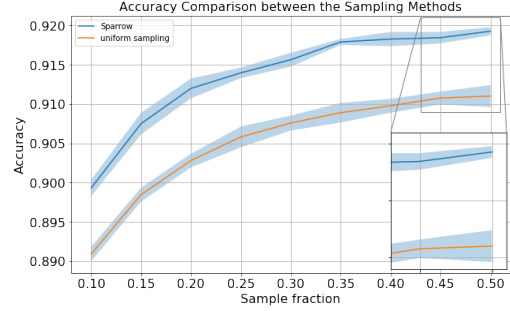

Figure 3: Accuracy comparison on the Cover-Type dataset. For uniform sampling, we trained XGBoost on a uniformly sampled dataset with the same sample fraction set in Sparrow. The accuracy is evaluated with same number of boosting iterations.

rules are usually larger than $\gamma_{t+1}$. The weak rules are then added to the strong rule with a weight corresponding to $\gamma_{t+1}$ (the lower bound for their true edges) to avoid over-estimation. Lastly, the value of $\gamma_{t+1}$ shrinks over time when there is no weak rule with the larger edge exists.

**Sampler**   Our assumption is that the entire training dataset does not fit into the main memory and is therefore stored in an external storage **(a)**. As boosting progresses, the weights of the examples become increasingly skewed, making the dataset in memory effectively smaller. To counteract that skew, Sampler prepares a *new* training set, in which all of the examples have equal weights, by using selective sampling. When the effective sample size $n_{\text{eff}}$ associated with the old training set becomes too small, the scanner stops using the old training set and starts using the new one[2].

The sampler uses selective sampling by which we mean that the probability of an example $(x, y)$ being added to the sample is proportional to its weight $w(x, y)$. Each added example is assigned an initial weight of 1. There are several known algorithms for selective sampling. The best-known one is rejection sampling in which a biased coin is flipped for each example. We use a method known as *minimal variance sampling* [13] because it produces less variation in the sampled set.

**Stratified Storage and Stratified Sampling**   The standard approach to sampling reads examples one at a time, calculates the weight of the example, and accepts the example into the memory with the probability proportional to its weight, otherwise rejects the example. Let the largest weight be $w_{\text{max}}$ and the average weight be $w_{\text{mean}}$, then the maximal rate at which examples are accepted is $w_{\text{mean}}/w_{\text{max}}$. If the weights are highly skewed, then this ratio can be arbitrarily small, which means that only a small fraction of the evaluated examples are then accepted. As evaluation is time-consuming, this process becomes a computation bottleneck.

We proposed a stratified-based sampling mechanism to address this issue (the right part of Figure 1). It applies incremental update to reduce the computational cost of making predictions with a large model, and uses a stratified data organization to reduce the rejection rate.

To implement incremental update we store for each example, whether it is on disk or in memory, the result of the latest update. Specifically, we store each training example in a tuple $(x, y, H_l, w_l)$, where $x, y$ are the feature vector and the label, $H_l$ is the last strong rule used to calculate the weight of the example, and $w_l$ is the weight last calculated. In this way both the scanner and sampler only calculate over the incremental changes to the model since the last time it was used to predict examples.

To reduce the rejection rate, we want the sampler to avoid reading examples that it will likely to reject. We organize examples in a stratified structure, where the stratum $k$ contains examples whose weights are in $[2^k, 2^{k+1})$. It limits the skew of the weights of the examples in each stratum so that $w_{\text{mean}}/w_{\text{max}} \leq \frac{1}{2}$. Besides, the sampler also maintains the (estimated) total weight of the examples in each stratum. It then associates a probability with each stratum by normalizing the total weights to 1.

To sample a new example, the sampler first samples the next stratum to read, then reads examples from the selected stratum until one of them is accepted. For each example, the sampler first updates its weight, then decides whether or not to accept this example, finally writes it back to the stratum it belongs to according to its updated weight. As a result, the reject rate is at most $1/2$, which greatly improves the speed of sampling.

Lastly, since the stratified structure contains all of the examples, it is managed mostly on disk, with a small in-memory buffer to speed up I/O operations.

## 6 Experiments

In this section we describe the experiment results of **Sparrow**. In all experiments, we use trees as weak rules. First, we use the forest cover type dataset [10] to evaluate the sampling effectiveness. It contains 581 K samples. We performed a 80/20 random split for training and testing.

Besides, we use two large datasets to evaluate the overall performance of Sparrow on large datasets. The first large dataset is the splice site dataset for detecting human acceptor splice site [18, 1]. We use the same training dataset of 50 M samples as in the other work, and validate the model on the testing data set of 4.6 M samples. The training dataset on disk takes over 39 GB in size. The second large dataset is the bathymetry dataset for detecting the human mislabeling in the bathymetry data [11]. We use a training dataset of 623M samples, and validate the model on the testing dataset of 83M samples. The training dataset takes 100 GB on disk. Both learning tasks are binary classification.

The experiments on large datasets are all conducted on EC2 instances with attached SSD storages from Amazon Web Services. We ran the evaluations on five different instance types with increasing memory capacities, ranging from 8 GB to 244 GB (for details see Appendix A).

### 6.1 Effectiveness of Weighted Sampling

We evaluate the effectiveness of weighted sampling by comparing it to uniform sampling. The comparison is over the model accuracy on the testing data when both trained for 500 boosting iteration on the cover type dataset. For both methods, we generate trees with depth 5 as weak rules. In uniform sampling, we first randomly sample from the training data with each sampling ratio, and use XGBoost to train the models. We evaluated the model performance on the sample ratios ranging from $0.1$ to $0.5$, and repeated each evaluation for 10 times. The results are showed in Figure 3. We can see that the accuracy of Sparrow is higher with the same number of boosting iteration and the same sampling ratio. In addition, the variance of the model accuracy is also smaller. It demonstrates that the weighted sampling method used in Sparrow is more effective and more stable than uniform sampling.

### 6.2 Training on Large Datasets

We compare Sparrow on the two large datasets, and use XGBoost and LightGBM for the baselines since they out-perform other boosting implementations [5, 12]. The comparison was done in terms of the reduction in the exponential loss, which is what boosting minimizes directly, and in terms of AUROC, which is often more relevant for practice. We include the data loading time in the reported training time.

There are two popular tree-growth algorithms: depth-wise and leaf-wise [17]. Both **Sparrow** and LightGBM grow trees leaf-wise. XGBoost uses the depth-wise method by default. In all experiments, we grow trees with at most $4$ leaves, or depth two. We choose to train smaller trees in these experiments since the training takes a very long time otherwise.

For XGBoost, we chose the approximate greedy algorithm which is its fastest training method. LightGBM supports using sampling in the training, which they called *Gradient-based One-Side Sampling* (GOSS). GOSS keeps a fixed percentage of the examples with large gradients, and randomly sample from the remaining examples. We selected GOSS as the tree construction algorithm for LightGBM. In addition, we also enabled the `two_round_loading` option in LightGBM to reduce its memory footprint.

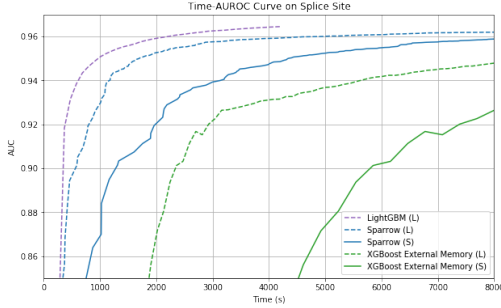
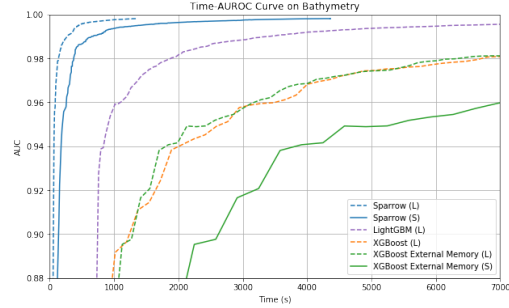

Figure 4: Time-AUROC curve on the splice site detection dataset, higher is better, clipped on right and bottom The (S) suffix is for training on 30.5 GB memory, and the (L) suffix is for training on 61 GB memory.

Figure 5: Time-AUROC curve on the bathymetry dataset, higher is better, clipped on right and bottom. The (S) suffix is for training on 61 GB memory, and the (L) suffix is for training on 244 GB memory.

The memory requirement of **Sparrow** is decided by the sample size, which is a configurable parameter. XGBoost supports external memory training when the memory is too small to fit the training dataset. The in-memory version of XGBoost is used for training whenever possible. If it runs out of memory, we trained the model using the external memory version of XGBoost instead. Unlike XGBoost, LightGBM does not support external memory execution. Lastly, all algorithms in this comparison optimize the exponential loss as defined in AdaBoost.

Due to the space limit, we put the detailed summary of the experiment results in Table 1 and Table 2 in the Appendix A. We evaluated each algorithm in terms of the AUROC as a function of training time on the testing dataset. The results are given in Figure 4 and Figure 5.

On the splice site dataset, **Sparrow** is able to run on the instances with as small as 8 GB memory. The external memory version of XGBoost can execute with a reasonable amount of memory (but still needs to be no smaller than 15 GB) but takes about 3x longer training time. However, we also noticed that **Sparrow** does not have an advantage over other two boosting implementations when the memory size is large enough to load in the entire training dataset.

On the bathymetry dataset, **Sparrow** consistently out-performs XGBoost and LightGBM, even when the memory size is larger than the dataset size. In extreme cases, we see that **Sparrow** takes 10x-20x shorter training time and achieves better accuracy. In addition, both LightGBM and the in-memory version of XGBoost crash when trained with less than 244 GB memory.

We observed that properly initializing the value of $\gamma$ and setting a reasonable sample set size can have a great impact on the performance of **Sparrow**. If stopping rule frequently fails to fire, it can introduce significant overhead to the training process. Specific to the boosted trees, one heuristic we find useful is to initialize $\gamma$ to the maximum advantage of the tree nodes in the previous tree. A more systematic approach for deciding $\gamma$ and the sample set size is left as future work.

## 7 Conclusion and Future Work

In this paper, we have proposed a boosting algorithm contains three techniques: *effective number of examples*, *weighted sampling*, and *early stopping*. Our preliminary results show that they can dramatically speed up boosting algorithms on large real-world datasets, especially when the data size exceeds the memory capacity. For future work, we are working on a parallelized version of **Sparrow** which uses a novel type of asynchronous communication protocol. It uses stopping rule to do the model update, and relaxes the necessity for frequent communication between multiple workers especially when training on large datasets, which we believe is a better parallel learning paradigm.

### Acknowledgements

We are grateful to David Sandwell and Brook Tozer for providing the bathymetry dataset.

This work was supported by the NIH (grant U19 NS107466).

## Footnotes

[1]The source code of the implementation is released at https://github.com/arapat/sparrow.

[2]The sampler and scanner can run in parallel on a multi-core machine, or run on two different machines. In our experiments, we keep them in one machine.

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
