[Supplementary Material]

Table 1: Training time (hours) on the splice site dataset. The (m) suffix is trained in memory. The (d) suffix is trained with disk as external memory.

Training time until the loss convergences

| Memory | Sparrow | XGB | LGM |
|---|---|---|---|
| 8 GB | **2.9** (d) | OOM | OOM |
| 15 GB | **8.4** (d) | > 50 (d) | OOM |
| 30 GB | **10.4** (d) | 0.6 (d) | OOM |
| 61 GB | 4.4 (d) | 12.8 (d) | **1.2** (m) |
| 244 GB | 1.3 (d) | 1.1 (m) | **0.5** (m) |
| Converged | 0.057 | 0.055 | **0.053** |

Training time until the average loss reaches 0.06

| Memory | Sparrow | XGB | LGM |
|---|---|---|---|
| 8 GB | **1.4** (d) | OOM | OOM |
| 15 GB | **7.1** (d) | > 50 (d) | OOM |
| 30 GB | **2.3** (d) | 9.3 (d) | OOM |
| 61 GB | 1.3 (d) | 4.6 (d) | **0.3** (m) |
| 244 GB | 0.5 (d) | 0.3 (m) | **0.2** (m) |

Table 2: Training time (hours) on the bathymetry dataset. The (m) suffix is trained in memory. The (d) suffix is trained with disk as external memory.

Training time until the loss convergences

| Memory | Sparrow | XGB | LGM |
|---|---|---|---|
| 8 GB | The disk cannot fit the data | | |
| 15 GB | **2.5** (d) | OOM | OOM |
| 30 GB | **1.9** (d) | 41.7 (d) | OOM |
| 61 GB | **1.2** (d) | 38.6 (d) | OOM |
| 244 GB | **0.4** (d) | 20.0 (m) | 4.0 (m) |
| Converged | **0.046** | 0.054 | 0.054 |

Training time until the average loss reaches 0.06

| Memory | Sparrow | XGB | LGM |
|---|---|---|---|
| 8 GB | The disk cannot fit the data | | |
| 15 GB | **1.0** (d) | OOM | OOM |
| 30 GB | **0.6** (d) | 41.7 (d) | OOM |
| 61 GB | **0.6** (d) | 38.4 (d) | OOM |
| 244 GB | **0.2** (d) | 16.9 (m) | 3.3 (m) |

## A  Evaluate Sparrow on Large Datasets

Due to the space limit, we summarize the detailed training time in each experiment in the appendix.

The experiments on large datasets are all conducted on EC2 instances with attached SSD storages from Amazon Web Services. We ran the evaluations on five different instance types with increasing memory capacities, specifically 8 GB (`c5d.xlarge`, costs $0.192 hourly), 15.25 GB (`i3.large`, costs $0.156 hourly), 30.5 GB (`i3.xlarge`, costs $0.312 hourly), 61 GB (`i3.2xlarge`, costs $0.624 hourly), and 244 GB (`i3.8xlarge`, costs $2.496 hourly).

In Table 1 and Table 2, we compared the training time it takes to reduce the exponential loss as evaluated on the testing data. Specifically, we compared the values of the average loss when the training converges and the corresponding training time. In addition, we observed that the average losses converge to slightly different values, because two of the algorithms in comparison, Sparrow and LightGBM, apply sampling methods during the training. Therefore, we also compared the training time it takes for each algorithm to reach the same threshold for the average loss.

We use "XGB" for XGBoost, and "LGM" for LightGBM in the tables. In addition, we observe that the training speed on the 8 GB instances is better than that on 15 GB instances, because the 8 GB instance has more CPU cores than the 15 GB instance.

## B  Stopping rule analysis

We set the stopping rule applied in Sparrow (Equation 8) based on the following theorem.

**Theorem 1 (based on Balsubramani [3] Theorem 4)** *Let $M_t$ be a martingale $M_t = \sum_i^t X_i$, and suppose there are constants $\{c_k\}_{k \geq 1}$ such that for all $i \geq 1$, $|X_i| \leq c_i$ w.p. 1. For $\forall \sigma > 0$, with probability at least $1 - \sigma$ we have*

$$\forall t : |M_t| \leq C \sqrt{\left( \sum_{i=1}^{t} c_i^2 \right) \left( \log \log \left( \frac{\sum_{i=1}^{t} c_i^2}{|M_t|} \right) + \log \frac{1}{\sigma} \right)},$$

*where $C$ is a universal constant.*

In our experiments, we set $C = 1$ and $\sigma = \frac{0.001}{|\mathcal{H}|}$, where $\mathcal{H}$ is the set of base classifiers (weak rules).

# C  Pseudocode for Sparrow

---

**Algorithm 1** Main Procedure of **Sparrow**

---

**Input** Sample size $n$
        A threshold $\theta$ for the minimum $n_{\text{eff}}/n$ ratio for training weak learner

**Initialize** $H_0 = 0$
**Create** initial sample $S$ by calling SAMPLE
**for** $k := 1 \ldots K$ **do**
    Call **Scanner** on sample $S$ generate weak rule $h_k, \gamma_k$
    $H_k \leftarrow H_{k-1} + \frac{1}{2} \log \frac{1/2+\gamma}{1/2-\gamma} h_k$
    **if** $n_{\text{eff}}/n < \theta$ **then**
        Receive a new sample $S \leftarrow$ from **Sampler**
        **Set** $S \leftarrow S'$
    **end if**
**end for**

---

**Algorithm 2** Scanner

---

**Input** An iterator over in-memory sampled set $S$
        Initial advantage target $\gamma_0 \in (0.0, 0.5)$

**static variable** $\gamma = \gamma_0$
**loop**
    **if** sample $S$ is scanned without firing stopping rule **then**
        Shrink $\gamma$ by $\gamma \leftarrow 0.9\widehat{\gamma}$
        Reset $S$ to scan from the beginning
    **end if**
    $(x, y, w_l) \leftarrow S.next()$
    $w \leftarrow$ UPDATEWEIGHT$(x, y, w_l, H)$
    **for** $h \in \mathcal{W}$ **do**
        Compute $h(\vec{x})y$
        Update $M_t, V_t$ (Eqn 7)
        **if** Stopping Rule (Eqn 8) fires **then**
            **return** $h, \gamma$
        **end if**
    **end for**
**end loop**

---

---

**Algorithm 3** Sampler

---

**Input** Randomly permuted, disk-resident training-set

Disk-resident stratified structure $D \leftarrow \{\}$
Weights of the strata $W \leftarrow \{\}$
Construct new sample $S \leftarrow \{\}$
**loop**
   With the probability proportional to $W$,
        **sample** a strata $R$
   $(x, y, w_l) \leftarrow R.next()$
   **Delete** $(x, y, w_l)$ from $R$, **update** $W$
   Receive new model $H$ from MAINPROCEDURE
   $w \leftarrow$ UPDATEWEIGHT$(x, y, w_l, H)$
   With the probability proportional to $w$,
        $S \leftarrow S + \{(x, y, w)\}.$
   **Append** $(x, y)$ to the right stratum with regard to $w$,
        $D \leftarrow D + \{(x, y, w)\}$
        **Update** $W$
   **if** $S$ is full **then**
      **Send** $S$ to MAINPROCEDURE
      $S \leftarrow \{\}$
   **end if**
**end loop**

---