[Reviews · NeurIPS 2019]

Reviewer 1



originality: Somehow novel. This paper uses "effective number of examples" and "weighted sampling", to reduce the used samples in each boosting round. quality: The idea is intuitive and seems can work. The author provides theoretical analysis and explicit experiments to check the performance of the proposed method. clarity: This paper is well written and organized overall. But the abstract is harsh. It is unclear what's the core idea and intuition of the paper from the abstract. It simply names the three techniques. significance: somehow significantly. The experiments show that Sparrow reduces the memory needed to train boosting trees, and in some cases converges faster than other baselines trained in memory. However, two benchmark datasets tree_size=4 make the results less attractive. cons and questions: - It is claimed that the inverse of "effective number of examples", i.e. the "variance of the estimator" is the standard quantifier for the accuracy of the empirical edge. However, there's no theory or referring to previous work supporting this claim. It would be better if the authors can provide at least the intuition behind this claim. - For experiments, the tree size is restricted to 4 leaves. Some datasets, especially large ones, would require much larger trees to get best result. Failing to train large trees faster makes boosting with external storage less attractive. - Stochastic Gradient Boosting (SGB), which randomly uses sub-samples and sub-features in each boosting round. I think this can reduce the memory cost for in-disk learning. Why not include it as the baseline? - Besides of the number of samples, do you consider to reduce the number of features in each boosting round? What is the hardness of this method? - I think only two benchmark datasets isn't sufficient to support the efficiency of Sparrow. - Does the time cost include data loading time? - As far as I know, the histogram algorithm in LightGBM isn't memory costly, and LightGBM has a parameter named tworoundloadg, that can use reduce the peak memory cost for large dataset. But from the supplementary, it seems this isn't enabled. It would be better to have the results with this setting. - Could you report the result of LightGBM without GOSS? From the LightGBM paper, the speed-up of GOSS is limited and hurt the accuracy a little bit. - The early stopping rule is based on a previous theorem (Theorem 1 in appendix B). However, the theorem requires the M_t to be a martingale. It is unclear how this condition is satisfied by the "difference between the true correlation of h and \gamma". Does it require the expectation of h(x)y to be exactly \gamma ? - In the weighted sampling experiment, only XGBoost is compared. As is stated in the paper, LightGBM provides another weighted sampling algorithm GOSS. Could you explain why GOSS is not used as baseline for evaluating the sampling method? - In appendix A, Sparrow does not scale well as the memory increases. Larger memory does not guarantee faster training. - The subscript i is missing in eqn (7). - Is the results run in SSD? if not, do you have results over SSD?

Reviewer 2



Update: I read authors' response and would like to keep my accept rating Summary: Authors look at boosting from a different angle that is commonly done. Each new step of boosting tries to build a learner that has highest correlation (they call it an edge) with a modified distribution of the instances labels. The edge is usually an empirical edge, calculated over the available sample of the data. Authors argue that such empirical edge is not always a good unbiased estimate of the true edge, which we really ultimately want to maximize. Then they argue that when memory is limited, we should only store examples with higher weights, since they contribute more to the estimate. So they propose to use weighted sampling to re-populate the sample with more important instances They also draw examples one by one and estimate the loss, once enough examples were drawn to give a good estimate, the best split is chosen. This leads to potentially smaller samples (we don't need to see the whole dataset to decide) and less memory consumption (keep only important samples). The experimental results show that at least on one dataset, the speedup AND generalization performance is much better than that of Xgboost and LightGbm I found it an interesting and a refreshing way of thinking about boosting. The paper is well written (albeit somewhat loaded, see my suggestions). My main critique is limited number of experiments (2 datasets) which produced mixed results. Comments/Questions: 1)What authors refer to "early stopping" - eg read instances until you saw enough to choose the best split is not new per se. For example Tensor Forest https://docs.google.com/viewer?a=v&pid=sites&srcid=ZGVmYXVsdGRvbWFpbnxtbHN5c25pcHMyMDE2fGd4OjFlNTRiOWU2OGM2YzA4MjE uses hoeffding bound. But i do admit that i have not seen it applied to boosting before, and the formulation is different and interesting 2) What happened in Splice dataset? What gave an edge to LightGBM there. 3) I think you need to run experiments on more datasets - among two tested, one is impressive and another is not so much Minor:Line 37 we flushes->we flush

Reviewer 3



This paper focuses on memory efficiency of boosting methods and proposes a new implementation of boosted trees. By combining three techniques, i.e., effective sample size, early stopping, and stratified sampling, the proposed method reduces the number of examples scanned while controlling the estimation error. Empirical results show that the proposed method can run efficiently with memory smaller than the training set and time faster than XGBoost and LightGBM. Here are several concerns and questions: 1. The paper should rearrange the algorithm description to make it more clear so that the algorithm can be reproduced. 2. The paper is well organized, while it needs to give the intuition of the main ideas in the early part which will make the paper more readable. 3. This paper uses a class imbalance example to explain the intuition of weighted sampling which is confusing. Besides, the two datasets seem to be imbalanced. In consideration of imbalanced classification problem, there are a number of methods reducing computation complexity. Related works about class-imbalance learning should be discussed. LightGBM also has options for imbalanced classification problem which should be discussed. Updated after reading feedback: I have read the author's feedback and all reviews. The feedback addresses my concerns, and I tend to vote for accepting this paper.

[Author Response · NeurIPS 2019]

We thank the reviewers for their feedback. We answer the questions from the reviewers in the order as they listed in the reviewer comments. Due to the space limit, we refer to the citations from the References in the original submission.

**Reviewer 1**

1. We apologize for not stating the intuition. Roughly speaking, let $Z = \sum_i^n w_i$ be the sum of the weights of the $n$ training examples, and let the true edge of a weak rule $h$ be $\gamma$. Then the expected (normalized) correlation between the predictions of $h$ and the true labels is $E\left(\frac{w_i}{Z} y_i h(x_i)\right) = 2\gamma$. The variance of this correlation can be written as

$$\mathrm{Var}\left[\frac{w_i}{Z} y_i h(x_i)\right] = \frac{1}{n^2} \frac{E\left(w_i^2\right)}{E^2(w_i)} - 4\gamma^2,$$

which is very similar to the form of the "effective number of examples" given in Equation 6.

2. We experimented with the deeper trees (of depth 5) on the cover type dataset (Section 5.1). Since it takes too long to get the training time till convergence on deeper trees on large datasets, especially for XGBoost, we limited the tree depth to 2 on the splice and the bathymetry datasets.

3. Thanks for the suggestion! We will consider it in our future work.

4. In this paper, we focus on the training data size in memory. Using different subset of the features in each boosting round will not reduce the memory footprint.

5. Thanks for the suggestion!

6. Yes, all running time includes the data loading time, which we believe is a practical way of evaluation in the context that the training data mostly resides on disk but cannot fit in the memory.

7. LightGBM uses the histogram-based method for training [12]. In addition, we enabled the "two_round_loading" parameter in all experiments presented in the paper. We will clarify it in the final version of the paper.

8. Removing GOSS will increase the amount of memory needed by LightGBM. We decided to keep it since the memory footprint is the main thing we are trying to optimize.

9. The relationship between martingales, sequential analysis, and stopping rules is somewhat involved [18]. Briefly, when the advantage of a rule is smaller than $\gamma$, then the sequence is a supermartingale. If it is larger than $\gamma$, then it is a submartingale. **The only assumption is that the examples are sampled i.i.d.**. Theorem 1 guarantees two things about the stopping rule defined in Equation 8: (1) if the advantage is smaller than $\gamma$, the stopping rule will never fire (with high probability); (2) if the stopping rule fires, the advantage of the rule $h$ is larger than $\gamma$.

10. At the time of writing the paper, we cannot successfully train LightGBM with GOSS on the cover type dataset because the software crashes for some reason that is unclear to us.

11. We believe the reviewer is referring to the first three rows of the top part of Table 1. Specifically, the top three training time of Sparrow. The explanation for the training times on the 8GB and 16GB instances is because the 16GB instance has fewer CPU cores than the 8GB instance. The difference between the 16GB and 32GB remains a mystery.

12. Yes, we ran all experiments on the AWS instances with an attached SSD storage.

**Reviewer 2**

1. Thank you for the reference! Indeed sequential analysis and stopping rules have a long history in statistics [18]. As discussed in the Related Work section, the "early stopping" technique has been investigated before (e.g. [4]), but they haven't considered using sampling to reduce memory footprint.

2. Our emphasis is on small memory sizes. LightGBM is sometimes faster when the memory size is *large enough to hold all training examples*, though the specific reason for why it is faster on the splice dataset is unclear to us.

3. Thanks for the suggestions!

Thanks for the improvement suggestions! We will address them to the best of our ability in our final version.

**Reviewer 3**

1. Thanks for the suggestion! We will try to make the explanation clearer.

2. We are using class imbalance (in Section 3.2) as one example in which resampling is beneficial. However, resampling is beneficial in a much wider set of situations, specifically, whenever the effective size of the sample is small. For example, this can happen when most of the examples are easy (have high margins).

Among the two big datasets we used in the experiment, the bathymetry dataset is balanced.

[Meta-Review · NeurIPS 2019]

All reviewers found this paper to have an interesting and novel take on boosting. Some found it extremely interesting.